# Health equity and wellbeing among older people's caregivers in New Zealand during COVID-19: Protocol for a qualitative study

Vanessa Burholt[1,2,3]*, Deborah Balmer[1], Rosemary Frey[1], Pare Meha[4], John Parsons[1], Mary Roberts[5], Mary Louisa Simpson[6], Janine Wiles[2], Moema Gregorzewski[7], Teuila Percival[5], Rangimahora Reddy[4]

1 School of Nursing, Faculty of Medical and Health Sciences, University of Auckland, Grafton, Auckland, New Zealand, 2 School of Population Health, Faculty of Medical and Health Sciences, University of Auckland, Grafton, Auckland, New Zealand, 3 Centre for Innovative Ageing, College of Human and Health Sciences, Swansea University, Wales, United Kingdom, 4 Rauawaawa Kaumātua Charitable Trust, Hamilton, Waikato, New Zealand, 5 Moana Research, Māngere, Auckland, New Zealand, 6 Waikato Management School, University of Waikato, Hamilton, New Zealand, 7 Centre for Arts and Social Transformation, Faculty of Education and Social Work, University of Auckland, Epsom, Auckland, New Zealand

☯ These authors contributed equally to this work.
* Vanessa.burholt@auckland.ac.nz

## Abstract

### Background

Knowledge of the challenges unpaid caregivers faced providing care to older people during the COVID-19 pandemic is limited. Challenges may be especially pronounced for those experiencing inequitable access to health and social care. This participatory action research study, located in New Zealand, has four main objectives, (i) to understand the challenges and rewards associated with caregiving to older care recipients during the COVID-19 pandemic restrictions; (ii) to map and collate resources developed (or mobilised) by organisations during the pandemic; (iii) to co-produce policy recommendations, identify useful caregiver resources and practices, prioritise unmet needs (challenges); and, (iv) to use project results in knowledge translation, in order to improve caregivers access to resources, and raise the profile and recognition of caregivers contribution to society.

### Methods and analysis

Māori, Pacific and rural-dwelling caregivers to 30 older care-recipients, and 30 representatives from organisations supporting caregivers in New Zealand will be interviewed. Combining data from the interviews and caregivers letters (from an archive of older people's pandemic experiences), framework analysis will be used to examine the interrelated systems of the human ecological model and the impact on caregiving experiences during the pandemic. Resources that service providers had created or used for caregivers and older people will be collated and categorised. Through co-production with caregivers and community partners we will produce three short films describing caregivers' pandemic experiences;

analysis included in participant information forms) and sensitivity (i.e. human data, Māori data sovereignty). These restrictions have been ratified by the Auckland Health Research Ethics Committee (AHREC) at the University of Auckland. Data will be available from AHREC (ahrec@auckland.ac.nz) on reasonable request. Data will be located in a controlled access repository at the University of Auckland.

**Funding:** This work was supported by Health Research Council of New Zealand (https://www.hrc.govt.nz/) Project Grant 20/1380 to Vanessa Burholt. The funders had and will not have a role in study design, data collection and analysis, decision to publish, or preparation of the manuscript.

**Competing interests:** The authors have declared that no competing interest exist.

identify a suite of resources for caregivers to use in future events requiring self-isolation, and in everyday life; and generate ideas to address unresolved issues.

## Introduction

Health equity has been defined as "the state in which everyone has the opportunity to obtain full health potential and no one is disadvantaged from achieving this potential because of social position or any other socially defined circumstances" [1]. The Ministry of Health in New Zealand recognises that people have differences in health that are not only avoidable but unfair and unjust [2]. The underlying causes of health inequity are largely located outside of the typically constituted domain of "health". However, the end result is a disproportionate burden of ill health in populations that face social, economic, cultural and political inequities. Equity recognises different people with different levels of advantage require different approaches and resources to produce equitable health outcomes [2].

In 2019, a cluster of people with pneumonia of unknown aetiology were identified in Wuhan, China [3] and by January 30 the World Health Organization declared the coronavirus disease 2019 (COVID-19) outbreak as a Public Health Emergency of International Concern [4]. In response to the COVID-19 pandemic, in April 2020 the Government of New Zealand (NZ) introduced a four-level alert system. At Alert Level 2 and higher, the Government recommended that people at higher risk of mortality from contracting COVID-19 should stay at home. Those at 'high risk' included older people (70+ years, or younger for Māori [the indigenous people of NZ] and Pacific peoples) and those with underlying health conditions [5] which are prevalent in the older population. During the pandemic, some older people in NZ will have spent more than 200 days confined to their home. In many cases, unpaid kin and non-kin caregivers supported older people by providing care and assistance with activities of daily living and home health care. In this study we use the term 'unpaid caregiver' to describe someone who provides care and support primarily because of a personal kin or non-kin relationship with the care-recipient. We do not use the phrases 'informal care' 'care partner' or 'carer'. The former contributes to the invisibility of unpaid caregivers in society as it conveys the impression that roles are unofficial (i.e. not defined by a salary, title or duties) and 'not essential' [6], while the latter do not clearly distinguish unpaid caregivers from paid care providers.

Caregiving in the community is often under-appreciated [7]. However, global health and social care systems are so reliant on unpaid caregivers that they constitute an essential workforce [8, 9]. In the international response to control COVID-19 transmission, unpaid caregiving has, and continues to be, the 'backbone' of health and social care provision in the community. However, the voices and stories of family caregivers have been invisible, and our knowledge of the successes, rewards, and challenges faced by those providing care to older people is limited. To date, articles assessing the situation of caregivers during COVID-19 have tended to focus on individual psychosocial outcomes such as depression [10], stress [11], burden [12, 13], suicidal tendencies [14], substance abuse [14], and psychological distress [15].

This research intends to raise the profile of unpaid caregivers providing essential support to older people, including those living with dementia, by recognising their significant social role during the COVID-19 pandemic in NZ. The research will generate new knowledge by exploring how family caregiving roles, access and use of resources changed. It will identify positive aspects of caregiving alongside unmet needs and challenges. The research will put knowledge into action by collating and co-designing resources for family caregivers to support their care

of older people. Pooling knowledge about caregiving during COVID-19 with international study teams (e.g. Southwest Health Equity Research Collaborative in Northern Arizona University), regarding caregiving in indigenous and minority populations, in the context of rurality and caregiving for people living with dementia, will facilitate cross-national knowledge transfer. Mapping responses in different countries aimed at supporting caregivers during the COVID-19 restrictions will help us identify policy, programmes or resources with the potential to improve national public health strategies and preparedness for future emergency responses to achieve the best possible outcomes for caregivers.

## Unpaid caregiving to older care recipients in NZ pre-COVID-19: The scale of the issue

Estimates suggest that 480,000 people (in a population of approximately 5 million) provide regular care for someone with an illness or disability [16]. This figure underrepresents spouses, Māori and Pacific caregivers [17].

Caregivers in NZ are most likely to be between 50–54 years, and provide on average 30 hours or more per week of care [18] for 8 years and 10 months [19]. However, significant numbers of younger people, older people, and adult children provide care to older people [20], for example, 34% of caregivers who provide care for household members with illness/disabilities are aged 55+ years [21]. Care may be provided to an older person by a number of caregivers [22] and 65% of caregivers are simultaneously in paid employment [16]. Caregivers' ethnicity reflects the population in general, with Māori and European New Zealanders most likely to report providing care. European New Zealander caregivers are more likely to report providing support to someone living in a separate household, than caregivers of other ethnicities who are more likely to report co-residence [17].

Caregiving for older people living with dementia is a major public health issue for societies worldwide. It is estimated that there are presently 70,000 people living with dementia in NZ, and the number is predicted to increase to 170,212 by 2050 [23]. In 2020, almost 87% of those with dementia in New Zealand were of European ethnicity, with Maori and Pacific people accounting for 6.2% and 2.8% of the population of people living with dementia respectively. *Dementia* is the largest single contributor to disability and need for care among older adults [24]. Globally, the vast majority of support and care for people living with dementia is provided through unpaid kin/non-kin networks (including spouses, adult children, children-in-law, and friends) [25].

In 2016, there were approximately 40,000 unpaid caregivers providing 45 million hours of care annually to people living with dementia in NZ, with an estimated cost of NZ$68.6 million [23]. Under normal circumstances, the provision of care and support to people living with dementia in the community can result in significant physical, emotional and economic strain for caregivers [26–32]. While we expect some features of caregiving to be similar regardless of the functional or cognitive characteristics of the person being cared for during the COVID-19 pandemic, we also anticipate that caregivers to people living with dementia faced some unique challenges (e.g. disruption of basic routines that promote mental health, and the unavailability of memory clinics and respite care [33]).

## Multilevel influences on caregiving pre COVID-19: The breadth of the issue

Pre-COVID-19 pandemic research identified several layers of influence on the caregiving experience. Within a human ecological systems framework, outcomes for caregivers and care recipients are influenced by social structural (public policy) and cultural norms, community,

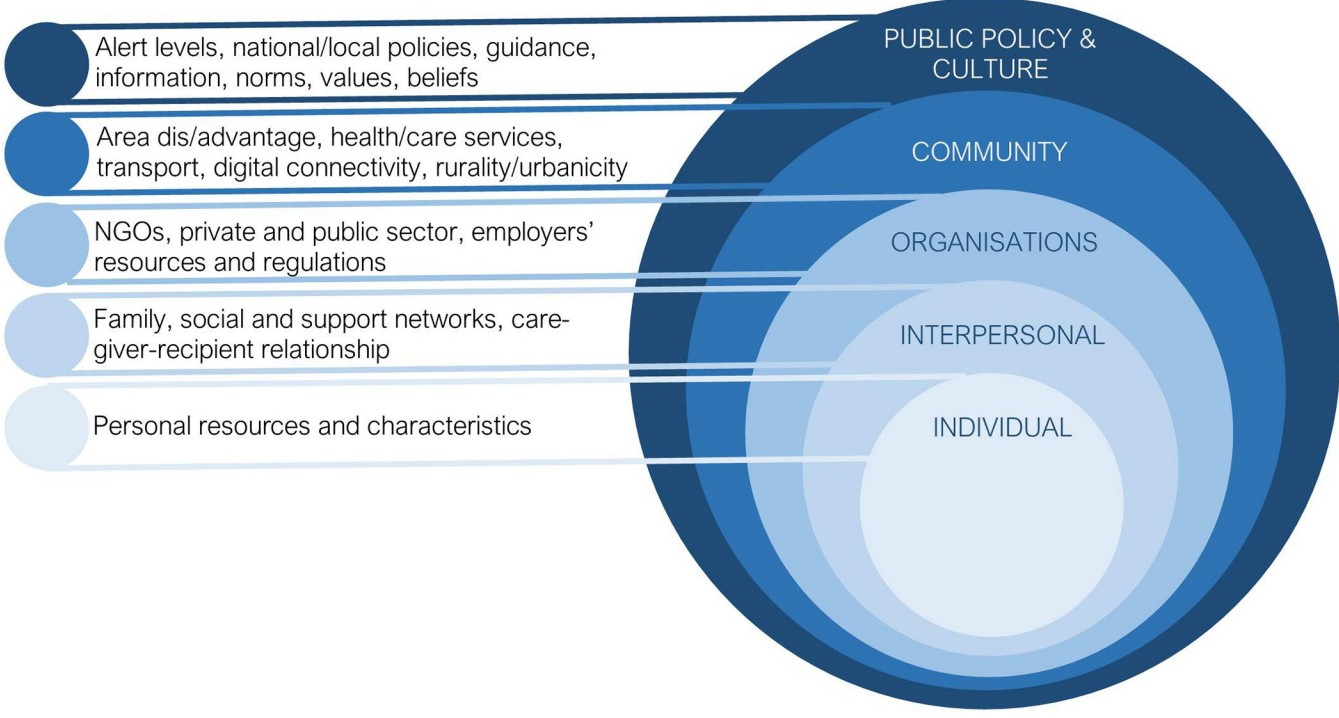

**Fig 1. Human ecological model comprising inter-related systems that influence caregiving.**

institutional/organisational factors, interpersonal processes and intrapersonal factors and the interaction between these factors (Fig 1).

**Social structural and cultural influences on caregiving.** The NZ healthcare system depends heavily on caregivers to provide support for older people [34]. Although government policies draw attention to the need to recognise the important role of family caregivers [35, 36] there remains little acknowledgment of their critical contributions to society [37–39]. National and local policies are responsible for eligibility and allocation of resources to caregivers/recipients, but bureaucratic obstacles [40] and difficulties with navigating health and social care systems present barriers to accessing services [41–43]. Some policies suggest that apps, online resources and social media may facilitate caregivers' connections with service and information, especially in isolated communities [35, 44], while others acknowledge that caregivers may need navigational help to receive the support and information they need [36]. The current Carers' Strategy Action Plan 2019–2023, *Mahi Aroha*, states that caregivers should be supported to continue in employment and to fully participate in a social life while providing support [36]. Despite these policy commitments, there is evidence of a strong direct relationship between caregivers' increasing sense of burden and discontinuation of their caring role [45].

In addition to public policy, there are cultural influences on caregiving. The term 'caregiver' may not resonate with everyone and there are cultural differences in usage [46]. Obligations to provide care that are based on normative expectations of family solidarity (prevalent in Pacific and Māori communities) may not be referred to as 'caregiving', and is an important reason why caregivers do not seek out services [47].

For Pacific peoples, older people are revered and caring for them is a prioritised service to one's family [48, 49]. Samoan people, for example, believe that caring for elders is an obligation and in return children will gain blessings (manuia).

For Māori, family (whānau) care of older people (kaumātua) is founded on commitment to reciprocal relationships and obligations to help [50] where older people are "carriers of culture, anchors for families, models for lifestyle, bridges to the future, guardians of heritage, and role models for younger generations" [51]. Family care is perceived as "normal" [52–54] and critical to the wellbeing of older people in the context of a significant ageing population and health inequities for Māori [55–59]. In caring for older people with dementia, family are crucial in promoting a healthy spirit or soul (wairua) [53]. The collective obligations of the family are informed by concepts such as love (aroha), showing respect, generosity and care for others (manaakitanga), genealogy (whakapapa) and relationships building through shared experiences and working together which provides a sense of belonging (whanaungatanga) [53]. Māori family caregiving sits at the heart of the wellbeing approach to care work (mahi aroha) integrating cultural identity, sustainability of the Māori world view (Te Ao Māori), social connections, knowledge and skills, human resource potential, and income [36].

Very little is known about the support needs of Māori family-based care networks, or Pacific care networks although caregivers have multiple demands on their time [60–62]. Pacific [63, 64] and Māori caregivers [53] are among those most in need of support, but are less likely to be aware of available services [47] and more likely to have lower levels of health literacy than European New Zealanders [65–67]. In this context, health literacy refers to the 'degree to which individuals have the capacity to obtain, process and understand basic health information and services needed to make appropriate health decisions' [63]. The additional challenge of English as a second language creates barriers in accessing health care services [63]. Both Māori and Pacific peoples have reported discrimination in health care services [68]. It has been suggested that health care providers need to work on their cultural competency, communicating information that resonates with the values and contexts of Māori and Pacific people, and developing appropriate outreach mechanisms to identify and connect with caregivers from different cultural groups [69–71].

**Community/Neighbourhood influences on caregiving.** In NZ, caregivers face significant challenges according to their geographic location or area deprivation [72]. The remoteness of communities affects access to health, social care and other essential (e.g. groceries) or community services [44, 73]. People living in remote rural areas are less likely to have a landline or cell phone coverage [44, 74]. Furthermore, area disadvantage may impact negatively on health outcomes for older caregivers [75].

Neighbourhoods have the potential to positively shape health, quality of life, and well-being for caregivers through the provision of local services, a sense of attachment to place [76, 77] and social interactions in 'third spaces' (e.g. coffee shops, parks and libraries) [78, 79]. In places with fewer opportunities for community participation and lower levels of social capital, caregivers of people living with dementia are more likely to report higher levels of isolation and find it harder to manage the demands of care [80].

In rural communities, there may be increased emphasis on family support, because of reduced options for alternative sources of support [81]. However, rural communities may be subject to ongoing population change and instability, and sources of formal or informal support may be stretched thinly [82–84]. Additionally, there may be an increased risk of abusive caregiver-recipient relationships, as there can be limited contact with people who could identify abuse or offer support in rural communities [81].

**Organisational influences on caregiving.** Unpaid caregivers perform an essential function as 'care-coordinators' linking older care-recipients to health care services, including primary health care and general practice. Caregivers attempt to meet gaps in formal health and social services for older adults [8, 85–91], advocating for care recipients, [43, 85, 92–94], facilitating access to, and explaining health information [86, 95]. High-quality care at home relies

upon strong relationships between health professionals and caregivers, good communication [62, 96, 97], and collaborative working [35, 98]. In NZ, services and support for caregivers is provided through primary care, District Health Board funded community services (e.g. Needs Assessment Support Coordination) and non-governmental organisations [43, 99] and the Supported Living Payment. Despite some recent effective cross-sectoral coordination [43, 61, 62, 88–90] the uptake of caregiver support is low, indicating a lack of flexibility [43, 100], inadequate provision [43], lack of knowledge about services [101], and a disconnection between what is offered and what is needed [102]. Unmet needs for formal services places increased demands on caregivers [103] and organisations may provide services that are neither relevant to caregivers needs and consistent with their wishes [40], nor socially, culturally or linguistically appropriate and accessible.

For working caregivers, the organisational characteristics of employers are also important. Juggling work and caregiving roles can impact negatively on health and well-being, especially where employers do not have caregiver support strategies or have inflexible working policies [104, 105]. The Employment Relations Flexible Working Arrangement Act 2007/2008 allows employees to request flexible working arrangements. However, fewer than one-third of workers are aware of this and many use annual or sick leave to provide care [106]. Furthermore, many caregivers reduce working hours or exit the workforce in order to provide care to an older person [104, 107], often negatively influencing their financial resources [108].

**Family, social networks, interpersonal relationships influence on caregiving.** Unpaid care-networks of older people may comprise partners, family networks, or mixed networks (care from children, other family members and professionals) [109–113] and the quality and extent of support are critical to caregivers' experiences. Particular caregiver-recipient relationships may produce different outcomes, for example, spousal caregivers have worse mental health outcomes than adult-child caregivers [114]. Furthermore, not all caregiver-recipient relationships are positive: conflict, tension and negative interactions are salient [115, 116]. Kin-networks may be disrupted by divorce and remarriage which may alter intergenerational bonds and responsibilities. Negative interactions within stepfamilies or discordant coordination of care within families can lead to greater levels of depression and caregiver burden [115, 117]. The risk of abusive relationships increases when individuals, families are isolated, or when caregivers do not receive appropriate support (e.g. coping with particular challenges associated with dementia) [17].

**Individual characteristics and resources influence on caregiving.** A large body of global research emphasises negative impacts of caregiving on physical, psychosocial, emotional, social and financial circumstances, otherwise known as 'caregiver burden' [118]. In terms of health equity, the risks for poor mental and physical health associated with caregiving have been highlighted [119–121], especially for caregivers of people living with dementia [26–31]. In NZ, caregivers are more likely to experience poor health [108] depression and anxiety than non-caregivers [122]. Research suggests that female caregivers and Māori caregivers have the poorest mental health [19] suggesting that these are the downstream corollaries of social, economic, cultural and political inequities.

Elsewhere, research has demonstrated that caregiving elicits a high economic burden [20]. However, little is known about the costs incurred by family caregivers in NZ [22]. Caregivers may experience out-of-pocket expenditure on services (e.g. personal care for care recipient or respite care), goods (e.g. bedding, mobility aids and incontinence products), home adaptations to improve accessibility, increased heating, laundry and transportation fees [123], but may also incur a loss of income associated with changes to employment [20]. Caregivers in NZ have lower economic living standards compared to non-caregivers [108]. An inadequate income

increases the risk of poor health and depressive symptoms for caregivers [124], and can restrict access to private care services [40].

As identified above, much of the academic literature on caregiving to older people focuses on negative outcomes, implying that it is financially physically and psychologically harmful [119]. This has been referred to as the 'penalties of caring' [46]. However, it is also important to recognise positive aspects of caregiving that are often ignored or not assessed. Being able to provide care for a loved one is often highly valued [79, 125]. Emphasising positive aspects of the role can improve the sense of self-efficacy, feelings of accomplishment, the quality of the caregiver/care-recipient relationship and reinforce caregiver wellbeing [126].

## Unpaid caregiving to older care recipients during COVID-19: Our knowledge of the issue

Knowledge of the impact of COVID-19 on caregivers is emerging. For example, a survey by Carers NZ/NZ Carers Alliance provided an insight into the experiences of 676 caregivers during the COVID-19 pandemic [127]. The survey highlighted several issues including reduced access to services and essential supplies, concerns about paid care staff coming into contact with the care recipient [128], increased exhaustion and stress, and financial pressures. Although the survey provides an indication of some challenges, we are unable to distinguish between the concerns of caregivers providing support to older people or people living with dementia from others, or whether caregivers from different cultures/ethnicities, social or geographic locations experienced particular inequities in health and care provision. From our understanding of the factors that impacted on caregivers before the global pandemic, we can postulate that some of these will have been amplified during COVID-19 as the related measures (socially distancing/isolating) are likely to have been an additional contextual stressor [129].

At the policy level, while general announcements and updates were praised for keeping the population of NZ informed about key developments during the pandemic [127] to date, no research has assessed whether the information and resources addressed caregivers' needs, or whether there are unresolved issues that still need to be tackled in order for us to be prepared for future scenarios. For example, was guidance culturally safe and sensitive for Māori and Pacific caregivers [130]? Were there suitable support systems for caregivers who lived at a distance from care recipients? And were generic resources suitable for caregivers supporting people living with dementia?

At the organisational level we need to understand whether clinical home care service guidelines need updating in light of caregivers experiences during the pandemic. We also need to understand how employers supported working caregivers, and how this impacted on the caregiving experience.

At the community level, framing unpaid caregiving through families in terms of reciprocity, obligation, solidarity and love [131] is in danger of glossing over issues of social justice and inequity at the community level. Some communities could be described as impoverished care environments and may be least equipped to meet a such a social care agenda [132]. We need to identify inequities in health and social care support, and access to others services for caregivers in disadvantaged communities or living at a distance from 'resource centres' [133]. We also need to know more about exclusion from access to information due to a digital divide in access to broadband internet services [134] and whether caregivers living in communities that experienced unequal access before lockdown have been pushed further to the margins.

At the interpersonal level, many caregivers rely on the stability and connection of their family and kinship systems. We do not know the extent to which family and caregiving networks

were hampered or enabled by social distancing requirements during COVID-19. The caregiver-care recipient relationship may have also been strained due to increases in caregiving tasks, hyper-vigilance, and less respite.

At the individual level, in addition to the greater impact on personal resources, COVID-19 probably altered the daily life of employed caregivers [128, 135, 136]. In addition, some caregivers may have faced additional child-care pressures exacerbated by school and nursery closures. Research in the United Kingdom suggests that caregivers sought to reduce their own risk of infection, to protect older care recipients [128]. Consequently, caregivers employed in 'essential services' may have felt conflicted about undertaking work duties that could potentially put an older care-recipient at risk of infection. We need to understand the situation of caregivers (for older people with/out dementia), in terms of burden, physical and mental wellbeing, burnout [130], sleeplessness [137], social isolation and loneliness [138].

## A critical human ecological framework to examine caregiving during COVID-19

Globally, unpaid caregivers are a crucial human resource that improve health-care capacity especially in areas with inequitable/sub-optimal access to health and social care. A critical human ecological approach [139] provides a meta-framework for challenging health inequities and will be used to question taken-for-granted assumptions about caregivers during the COVID-19 pandemic. These assumptions included the continuing capacity of caregivers to provide care, their psychological-preparedness to provide support, their functional capacity to cope with demanding and physical caregiving tasks without respite [130], and their financial capacity to bear the brunt of additional caregiving costs. Additionally, there were implicit assumptions that caregiver-recipient relationships would be mainly co-resident dyads with little attention paid to the implication of social restrictions for non-co-resident caregivers, or caregiving that was distributed throughout families.

This research will examine the intersectionality of structural inequities, focusing on subgroups of caregivers who may face particular challenges, or may be excluded from existing policy [140], that is, those living in remote, rural disadvantaged areas [141], and Māori and Pacific caregivers. The critical human ecological framework has been selected to situate this research because this approach recognises that caregivers are embedded in different social or cultural groups, and that intersecting settings or levels of the model influence health equity and wellbeing. The health equity and wellbeing of caregivers will be conceptualised as an emergent product of a system, in which individual, community, environment and macro-structures (e.g. governmental policies, culture, values and normative beliefs) are inextricably connected [142].

While this research will highlight health inequities (in terms of outcomes for caregivers) that are the result of the intersectionality of structural inequities, the COVID-19 pandemic also provides a unique opportunity to maximise learning from any positive ways in which caregiving was disrupted. For example, opportunities to work at home [128], or to receive remote medical consultations (telemedicine), may have provided benefits that could or should be extended into life beyond the pandemic to improve quality of life. Taking a citizen-centred approach, the goal of this research is to provide the opportunity for caregivers in NZ to have a voice in shaping recommendations for future health and social policy, and services [143]. It will contribute to the development of solutions that are culturally responsive and relevant to NZ. Our international collaborations will provide the opportunity to identify experiences that were common to caregivers during the pandemic and point to vulnerabilities and areas that may require national and international reform to reduce the risk of future public health crises impacting on caregivers.

### Objectives

The objectives of this study are:

- To undertake interviews with caregivers, and select caregivers' letters from an extant archive, in order to understand the challenges and rewards associated with caregiving to older care recipients during the COVID-19 pandemic restrictions.

- To undertake interviews with organisations providing services to caregivers and older people, in order to map and collate resources developed (or mobilised) during the pandemic.

- To hold workshops and a co-design challenge event to co-produce policy recommendations, identify useful caregiver resources and practices, prioritise unmet needs (challenges).

- To use project results in knowledge translation, in order to improve caregivers access to resources, and raise the profile and recognition of caregivers contribution to society.

### Research questions

The research will address the following questions:

1. What were the experiences of Māori, Pacific and rural-dwelling caregivers (providing care to older people) in NZ during the COVID-19 pandemic?

2. What were the similarities and differences between the experiences of caregivers for people living with dementia compared to caregivers providing care for older people with other health and disability needs?

3. How did the interrelated systems of the human ecological model impact on caregiving experiences during the COVID-19 pandemic?

4. What can we learn from the experiences of caregivers and organisations supporting them, to prepare for other similar (emergency health) situations, or to transfer to new ways of working in everyday life?

## Methods

This study uses take a participatory action research approach [144, 145]. Phases 1–2 (concurrent qualitative studies) will underpin co-design activities in Phase 3 (transformative sequential phase). The co-production of knowledge is often referred to as integrated knowledge translation which describes an ongoing relationship between researchers, community groups and policy-makers for the purpose of engaging in a mutually beneficial research to support decision-making [146, 147]. Using participatory action research, integrated knowledge translation [148] will be achieved through a reflective cycle, whereby community partners collect and analyse data, and along with caregivers, identify what actions should follow.

Four main elements underpin our co-created research: (i) collaboration and equitable involvement in research; (ii) solidarity and capacity-building; (iii) empowerment and action for systems change; and (iv) sustainability [144]. Accordingly, all partners have actively contributed to the proposal, [144] in order to deliver customised research to improve local practice [149].

### Phase 1: Experiences of caregivers

**Sample and setting.**    Two samples will be used in Phase 1 of the study, extant data and new primary data.

1. Extant data for caregivers extracted from a study entitled *Social connectedness among older people during Covid-19* (Auckland Medical Research Foundation; Principal Investigator Professor Merryn Gott). This study requested letters from people age ≥ 70 years living in NZ describing their experiences of COVID-19. Data collected between May 2020 and May 2021 comprises >650 responses. We will select letters sent by older caregivers from the main dataset by searching files for key words ('care' 'support' and synonyms).

2. A new cross-sectional purposive critical case sample [150] of caregivers providing support to 15 older people living with dementia, and 15 without dementia (i.e. caregivers to five older people living with dementia and five without in each group: Māori, Pacific and rural). Established and trusted relationships are critical to conducting community research [151]. Participants will be volunteers identified through our collaborating community group networks who live and work within the communities.

**Study design.** In a cross-sectional qualitative study, caregivers who provide care or support to an older person, within the three groups/settings will be invited to take part in face-to-face guided interviews in their own homes, or in a setting of their choice. Individual, joint, or family-based interviews will be offered to participants. Consequently, the number of caregivers interviewed (providing care to 30 care-recipients) is likely to be greater than 30. Face-to-face interviews are preferred by Māori caregivers [123] and caregivers with mobility, hearing or vision difficulties who may encounter difficulties with written responses or telephone interviews [152, 153]. However, if the Alert Levels associated with the COVID-19 pandemic are in place and we are unable to interview face-to-face, we will adopt a contingency plan and conduct interviews by Zoom/Skype/ telephone [154].

A short questionnaire will be used to establish the demographic characteristics of the caregiver(s) (S1 Appendix, Demographic Questions: Caregivers). An interview guide will be aligned to the 'levels' of the human ecological model and will focus on the individual, interpersonal relationships, organisational issues, community and public policy and culture (S2 Appendix, Semi-Structured Interview Guide for Individual Caregivers). The first area of enquiry will focus on the individual and is structured to establish changes in caregiving during COVID-19 restrictions and positive and negative outcomes on personal resources and wellbeing. This section will also explore whether there were challenges balancing caregiving and additional roles (e.g. mother, employee or employer) during COVID-19 restrictions. The second section will explore family, social networks, interpersonal relationship in relation to caregiving. The third section, the organisational area of enquiry, will establish whether clinical home care service guidelines need updating in light of experiences during the pandemic. The topic guide will cover issues relating to encounters with home care providers during COVID-19, use of telecare and telehealth, alongside issues concerning the availability of resources (e.g. PPE, respite, day care), and the impact on caregivers. The fourth section, focusing on the community/neighbourhood area of enquiry, will identify inequities in access to services (including broadband and internet services) for caregivers in disadvantaged communities and for those living at a distance from resource centres. The fifth section will address the social structural and cultural area of enquiry, and will explore whether information and resources developed during the pandemic addressed caregivers' needs, or whether there are unresolved issues that still need to be tackled in order to be fully prepare for future scenarios. A modified interview guide will be used for group interviews with two or more caregivers: some questions will be directed at the main caregiver, while others will be asked of each participating caregiver (S3 Appendix, Semi-Structured Interview Guide for Groups [>1 Caregiver]).

Community Researchers working with Māori and Pacific caregivers will interview participants in Te Reo Māori (the Māori language), Samoan, Tongan, Niuean or English according to the caregivers' preferences. Face-to-face interviews will be informed by culturally aligned research ethics and processes. All participants will be offered NZ$50 voucher for their involvement in the research [155–157].

**Analysis.** A thematic analysis of secondary and primary cross-sectional qualitative data will be undertaken. Transcripts will be anonymised [158], imported and analysed using QSR NVivo Version 12. Drawing on social phenomenology [159] analysis will utilise both inductive and deductive thematic analysis to interpret the raw data within a framework analysis [160].

Familiarisation, conceptual and cultural understanding of the interviews will be clarified during team meetings. A first version of the coding index will comprise top level headings for themes heavily rooted in a priori issues (i.e. human ecological model). Two interviews from each group will be independently coded by the researchers during which categories (sub themes) will be identified and refined. The resulting index (comprising top level heading and clustered sub-themes) will be systematically applied to the text. Data will be charted into a framework to provide decontextualised descriptive accounts in relation to the different levels of the model and experiences during COVID-19 for each care-recipient (from the perspective of one or more caregiver). Schematic diagrams will be developed and used alongside the charted material to capture commonalities of experiences across caregivers [161, 162] and at each level of the model (e.g. identifying key issues relating to community disadvantage). Chronological explanatory summaries capturing the complexity of experiences will be created for each participant describing the interrelationship between levels of the model [163].

A Māori Expert Advisory Group and the collaborating Pacific Research Group will help guide interpretation of Māori and Pacific data analysis. Researchers will meet regularly to discuss different ways of interpreting the research phenomenon, differences between groups, and the persuasiveness of the analysis [164].

## Phase 2: Organisational support and resources for caregivers

**Sample and setting.** In a cross-sectional qualitative study, a purposive sample of 30 national and local organisations across NZ (including Māori and Pacific organisations) who provided support to caregivers during COVID-19 will be selected through collaborating community organisations.

**Study design.** CEOs/Directors of the service provider organisations will be contacted by phone/email to describe the purpose of the study. An appointment for a 30–45 minute telephone/Zoom interview will be made with the CEO/Directors or with a nominated member of the organisation. Guided interviews will follow a similar format to caregivers' interviews. The organisational representative will be asked for their perception of caregivers' successes and challenges during COVID-19. They will be asked to describe the organisational response to caregiver needs and asked to describe any resources/services that they developed specifically to address these (S4 Appendix, Semi-Structured Interview Guide for Service Providers).

**Analysis.** Telephone/Zoom interviews will not be recorded or transcribed verbatim. As these interviews do not require a particular closeness between researchers and the interviewees, we will use a reflexive, iterative process of data management comprising concurrent note-taking; reflective journaling immediately post-interview; summarising the description of resources, and undertaking preliminary content analysis identifying emerging themes [165].

The interviewee will be sent a (i) description of resources deployed/developed by the organisation and (ii) the major themes elicited in relation to the perception of caregivers (un)met needs. The participant will be asked to verify and/or amend the summaries to ensure

stakeholder engagement and 'member checking' [166, 167]. The themes will be added to the framework analysis (from Phase 1). The descriptions of resources will summarised into 'plain' language. For informational resources hard copies or URLs will be obtained.

## Phase 3: Co-produced resources, recommendations and priorities for caregivers

**Sample and setting.**   Caregivers who highlight an interest during Phase 1 will be eligible to participate in any of the co-creation workshops. We will organise six co-creation workshops (two in each setting) to assess the adequacy of collated caregiving resources and prioritisation of unresolved challenges. We anticipate a maximum of five participants in each of these workshops. We will convene one creative co-design challenge event [168]. Caregivers from Phase 1 that express an interest in telling their stories on film will be invited to take part in digital storytelling.

**Study design.**   We will convene six workshops (two each with Māori, Pacific and rural caregivers) for ideation concerning appropriate recommendations and resource materials/ models to support family caregivers. Each workshop will last about 2–3 hours. They will be conducted by researchers skilled in hosting co-production events, and will use cultural ways of working. The first event will cover the following issues:

- A summary of the key challenges faced by caregivers in each setting identified in Phase 1

- A summary of the resources deployed by organisations to meet caregivers' needs identified in Phase 2

- Discussion around whether resources met caregivers' key challenges during COVID-19

- Identifying good practice for future health disasters (pandemics) and emergency situations

- Identification of good practice that could be applied to everyday life (extended to beyond pandemic preparedness) to support caregivers

At the second event, the participants will discuss insights, ideas for new practices, services, policy recommendations: with particular attention given to identified unmet needs (identified at the first workshop event) for information, practical health and social care support, health literacy, and psychosocial help. Ideas generated for new resources would be prioritised by each group to ensure successful application of study findings.

In order to address any unresolved prioritised issues, a creative co-design challenge event would bring together stakeholders (e.g. caregivers from the workshops, policymakers, service providers, designers, technology companies) to address the challenges [168]. During the creative co-design challenge event, small teams would co-produce a brief and implementation plan to move each challenge forward beyond the project funding period. Results will inform the design of practice, policy and future collaborative studies to address areas of high priority as indicated by the caregivers.

Three short films (one each for Māori, Pacific and rural caregivers) would be co-created to communicate caregiver 'stories'. An experienced researcher and a filmmaker will work with three individuals or groups (e.g. a group of family caregivers) to construct and share caregivers' own authentic narratives. The caregivers' interview transcripts will be used as an aide-mémoire to create 'storyboards' which will serve as a visual layout for story construction. Each short film will comprise 5–6 minutes of visual narrative creating compelling accounts of caregiving experiences [169, 170].

## Ethics

Ethical approval was obtained from the Auckland Health Research Ethics Committee (reference AH21966) on the 31 May 2021.

## Results

Data collection (Phase 1 and 2) commenced in June 2021 and will be completed by September 2021. Analysis of data will feed into Phase 3 to develop caregiver resources, digital stories, recommendations for organisational good practice, policy and coordination.

An integrated knowledge transfer strategy [146–148] will maximise the impact of outputs from the research within and beyond the grant lifecycle. Co-production and working with community-based partners and caregivers will result in realistic real-world application and will facilitate a pathway to impact through implementation of recommendations in primary care and maximising caregiver access to support resources.

## Discussion

The need for this research was identified by our community partner groups. Subsequently, community groups and academics have risen to the challenge by jointly and collaboratively developing a proposal to identify and address the needs of specific sub-groups of caregivers who are likely to have been particularly affected by sub-optimal and inequitable access to advice, health and social care support and resources–that is Māori, Pacific and rural-dwelling caregivers. From the outset, the team has taken a participatory action research approach to project design which will continue throughout implementation to ensure translation of social research into public health benefit for family caregivers to older people ($\approx$480,000).

We anticipate that the research will have social, economic and cultural benefits. The research will improve caregivers' access to support resources, caregiving safety, effectiveness and cultural appropriateness during future emergency health situations. We will also identify good practice and resources developed during the pandemic that could be rolled out to positively impact on caregivers' everyday life. Furthermore, the research is anticipated to improve the responsiveness of health and social services to meet caregivers' needs, especially in populations that are vulnerable to exclusion and health inequity.

At the end of the study, resources will provide practical solutions for challenges associated with caregiving during and beyond the pandemic, especially for Māori, Pacific and caregivers living in remote, rural or disadvantaged areas. As shortfalls in support impact on caregiver burden, physical and cognitive health, quality of life, and other psychosocial outcomes (e.g. loneliness), we anticipate providing lasting societal benefits, potentially delaying institutionalisation of older care recipients, and reducing health and social care costs. The social, economic and cultural benefits will be achieved through the collation and distribution of best practice and support resources, recommendations for policy (especially the National Health Emergency Plan [171], in terms of readiness, recovery and response), and identifying unresolved challenges to be addressed beyond the funded duration of the project. Examining caregivers' experiences in more than one location with international partners will help to identify good practice, and perhaps clarify the drivers of suboptimal support, while deepening our knowledge about needs and expectations of caregivers. In turn, this could improve the quality of care, support and services, by broadening and strengthening evidence for change beyond the confines of a single nation.

## Supporting information

**S1 Appendix. Demographic questions: Caregivers.**
(PDF)

**S2 Appendix. Semi-structured interview guide for individual caregivers.**
(PDF)

**S3 Appendix. Semi-structured interview guide for groups (>1 caregiver).**
(PDF)

**S4 Appendix. Semi-structured interview guide for service providers.**
(PDF)

## Author Contributions

**Conceptualization:** Vanessa Burholt, Deborah Balmer, Rosemary Frey, John Parsons, Mary Louisa Simpson, Janine Wiles, Moema Gregorzewski, Teuila Percival, Rangimahora Reddy.

**Funding acquisition:** Vanessa Burholt.

**Investigation:** Vanessa Burholt, Deborah Balmer, Rosemary Frey, John Parsons, Mary Louisa Simpson, Janine Wiles, Moema Gregorzewski, Teuila Percival, Rangimahora Reddy.

**Methodology:** Vanessa Burholt, Deborah Balmer, Rosemary Frey, John Parsons, Mary Louisa Simpson, Janine Wiles, Moema Gregorzewski, Teuila Percival, Rangimahora Reddy.

**Project administration:** Vanessa Burholt.

**Writing – original draft:** Vanessa Burholt.

**Writing – review & editing:** Vanessa Burholt, Deborah Balmer, Rosemary Frey, Pare Meha, John Parsons, Mary Roberts, Mary Louisa Simpson, Janine Wiles, Moema Gregorzewski.

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
