## [Decision Letter · Decision Letter 0]

28 Apr 2022

PONE-D-21-26434

Health equity and wellbeing among older people’s caregivers in New Zealand during COVID-19: Protocol for a qualitative study

PLOS ONE

Dear Dr. Burholt,

Thank you for submitting your manuscript to PLOS ONE. After careful consideration, we feel that it has merit but does not fully meet PLOS ONE’s publication criteria as it currently stands. Therefore, we invite you to submit a revised version of the manuscript that addresses the points raised during the review process.

We look forward to receiving your revised manuscript.

Kind regards,

Maw Pin Tan, M.D.

Academic Editor

PLOS ONE

Reviewers' comments:

Reviewer's Responses to Questions

**Comments to the Author**

1. Does the manuscript provide a valid rationale for the proposed study, with clearly identified and justified research questions?

Reviewer #1: Yes

Reviewer #2: Yes

2. Is the protocol technically sound and planned in a manner that will lead to a meaningful outcome and allow testing the stated hypotheses?

Reviewer #1: Yes

Reviewer #2: Yes

3. Is the methodology feasible and described in sufficient detail to allow the work to be replicable?

Reviewer #1: Yes

Reviewer #2: Yes

4. Have the authors described where all data underlying the findings will be made available when the study is complete?

Reviewer #1: No

Reviewer #2: No

5. Is the manuscript presented in an intelligible fashion and written in standard English?

Reviewer #1: Yes

Reviewer #2: Yes

6. Review Comments to the Author

You may also provide optional suggestions and comments to authors that they might find helpful in planning their study.

Reviewer #1: This is excellent and much-needed research. I congratulate the authors.

I have only one comment. In lines 413 & 414, the authors have mentioned "15 older people living with dementia, and 15 without dementia" will be selected from a convenient sample. I understand that only five caregivers from each of the three categories will be included. I wonder if this is an adequate sample. Maybe the authors should have aimed for data saturation before restricting the selection to just five from each group.

Reviewer #2: This is a valuable piece of research - thank you for the invitation to review the protocol, which is very well written and generally very sound.

I only have two principal comments:

Equity is a multi-faceted term with a number of potentially different definitions, and 'Health equity' too. What is described here looks to be 'equity of potential access to services' (demand/need perspective), rather than 'equity of health outcomes', though the two are of course ultimately related, and the latter does make a sort of appearance at some point. It also appears to include 'equity of health service provision or delivery (supplier perspective)'. The term 'Health equity' is used in the title, and the concept of inequities is mentioned throughout - fairly infrequently to be honest, given the prominence of the term in the title. However, currently, no definition is provided. This should appear early in the piece and be sufficiently detail.

The choice of the Human ecological model - as opposed to other potentially relevant frameworks or models - needs a fuller and more detailed justification.

Other comments: Abstract has too much repetition, and needs rewriting; sampling for phase I needs clarifying if it is convenient or purposive.

I have attached the pdf with some comments, typos etc.

7. PLOS authors have the option to publish the peer review history of their article (what does this mean?). If published, this will include your full peer review and any attached files.

Reviewer #1: **Yes: **Sridhar Vaitheswaran

Reviewer #2: **Yes: **Christopher Carroll

---

## [Author Response · Author response to Decision Letter 0]

13 May 2022

Reviewer #1: This is excellent and much-needed research. I congratulate the authors.

I have only one comment. In lines 413 & 414, the authors have mentioned "15 older people living with dementia, and 15 without dementia" will be selected from a convenient sample. I understand that only five caregivers from each of the three categories will be included. I wonder if this is an adequate sample. Maybe the authors should have aimed for data saturation before restricting the selection to just five from each group.

We acknowledge that this is a limitation of our research that will be acknowledged in future publications of results from this study. Although we would have loved to have a larger sampler, the grant ceiling was NZ$250K and this was the maximum number of interviews that we could achieve (taking into account translation and transcription costs etc). 

Reviewer #2: 

I only have two principal comments:

Equity is a multi-faceted term with a number of potentially different definitions, and 'Health equity' too. What is described here looks to be 'equity of potential access to services' (demand/need perspective), rather than 'equity of health outcomes', though the two are of course ultimately related, and the latter does make a sort of appearance at some point. It also appears to include 'equity of health service provision or delivery (supplier perspective)'. The term 'Health equity' is used in the title, and the concept of inequities is mentioned throughout - fairly infrequently to be honest, given the prominence of the term in the title. However, currently, no definition is provided. This should appear early in the piece and be sufficiently detail.

NOTE TO EDITOR – the line numbers do not match between the tracked changes version and non-tracked version of the MS. We have not been able to resolve this issue (with a skip in numbering from 49-57 in the tracked version). The line numbers in the response to reviewers refer to the non-tracked changes version of the MS.

The following definition has been provided at the beginning of the protocol (Line 51): “Health equity has been defined as “the state in which everyone has the opportunity to obtain full health potential and no one is disadvantaged from achieving this potential because of social position or any other socially defined circumstances” [1]. The Ministry of Health in New Zealand recognises that people have differences in health that are not only avoidable but unfair and unjust. The underlying causes of health inequity are largely located outside of the typically constituted domain of “health”. However, the end result is a disproportionate burden of ill health in populations that face social, economic, cultural and political inequities. Equity recognises different people with different levels of advantage require different approaches and resources to produce equitable health outcomes.” We think that this now frames the rest of the proposal clearly delineating the difference between health equity as an ‘outcome’ and different pathways or causes of health inequity. 

Two new references have been added in this paragraph:

National Academies of Sciences Engineering and Medicine Health and Medicine Division, Health and Medicine Board on Population Health and Public Health Practice, Committee on Community-Based Solutions to Promote Health Equity in the United States, Baciu A, Negussie Y, Geller A, et al. Communities in Action: Pathways to Health Equity: National Academies Press; 2017.

Ministry of Health New Zealand Government. Achieving equity: Ministry of Health New Zealand Government; 2022 [cited 2022 11 May]. Available from: https://www.health.govt.nz/about-ministry/what-we-do/work-programme-2019-20/achieving-equity#:~:text=World%20Health%20Organization.-,The%20definition,to%20get%20equitable%20health%20outcomes

(Reviewer’s comment boxes)

Is this the reference to 'health equity'? Again, there are multiple definitions of the term - what is described here looks to be like 'equity of potential access to services' (demand/need perspective), rather than 'equity of health outcomes', though the two are of course ultimately related. 

The article now states that we are talking about health equity as outcomes that result from inequitable access to social, economic, cultural, political, health and social care resources, with a major focus on the latter (as relevant during the COVID 19 public health restrictions). For example, in the abstract we note: “Challenges may be especially pronounced for those experiencing inequitable access to health and social care”. 

Thereafter, we clearly refer to pathways to health inequities as distinct for health inequity as an outcome: 

• Line 297 “caregivers from different cultures/ethnicities, social or geographic locations experienced particular inequities in health and care provision”

• Line 338 “in danger of glossing over issues of social justice and inequity at the community level. Some communities could be described as impoverished care environments and may be least equipped to meet a such a social care agenda [130]. We need to identify inequities in health and social care support, and access to others services for caregivers in disadvantaged communities or living at a distance from ‘resource centres’”

• Line 343: “especially in areas with inequitable/sub-optimal access to health and social care”. 

• Line 464 “will identify inequities in access to services (including broadband and internet services)”

• Line 606 “particularly affected by sub-optimal and inequitable access to advice, health and social care support and resources”

• Line 175 “in the context of a significant ageing population and health inequities for Māori.”

Amended line 322 “caregivers living in communities that experienced unequal access before lockdown have been pushed further to the margins.”

Amended line 343 “A critical human ecological approach provides a meta-framework for challenging health inequities”

This is now moving towards health equity in terms of health outcomes. 

Line 261. In this section entitled Individual characteristics and resources influence on caregiving we note that “A large body of global research emphasises negative impacts of caregiving on physical, psychosocial, emotional, social and financial circumstances, otherwise known as ‘caregiver burden’ [118]”. Thus, ‘caregiver burden’ is not synonymous with health equity. Therefore, in this section we have highlighted ‘health equity’ in terms of health outcomes; we have also added a line to indicate that “these are the downstream corollaries of social, economic, cultural and political inequities.”

The choice of the Human ecological model - as opposed to other potentially relevant frameworks or models - needs a fuller and more detailed justification.

(Reviewer’s comment box)

Needs a reference at the end of the sentence - also the choice of this particular model needs more detailed justification. [Line 330]. 

Line 344, The reference for the ecological framework is in the middle of the sentence. The last part of the sentence is about the purpose of using this framework in the current study – this does not have a reference. 

We have added the following statement to justify the choice of the critical human ecological framework Line 356: “The critical human ecological framework has been selected to situate this research because this approach recognizes that individuals are embedded in different intersecting settings, and that social or cultural group experiences can shape how individuals experience these settings, and ultimately health equity. Embedding experiences within larger systems of culture (macrosystem) and time (chronosystem), will be used to explicitly acknowledge issues of power, oppression and privilege, for example in the experiences of access to health and social care during the COVID19 public health restrictions.”

Other comments: Abstract has too much repetition, and needs rewriting; sampling for phase I needs clarifying if it is convenient or purposive. 

(Reviewer’s comment boxes)

Deleted text describe the methods to be used, rather than the objectives, and is repeated below. Done.

Part ii - listed above - seems to be missing here. Abstract states, “and 30 representatives from organisations supporting caregivers in New Zealand will be interviewed”. Added the line “Resources that service providers had created or used for the caregivers and older people will be collated and categorised.”

Other comment boxes on pdf:

'unheard' rather than 'invisible'? Not amended. See references 20, 48, 49, 93 and 128 for the ‘invisible workforce’ that refers to the status of unpaid caregivers.

Are these 40,000 still unpaid caregivers? Or paid and unpaid? The reference given is not clear. Added ‘unpaid’ in text.

'caregivers' is unhyphenated elsewhere. Amended to caregivers throughout the manuscript.

Multiple potential definitions of this term - including, but limited to, being able to find health information, and being able to understand and action health information - maybe define for this context. Added to text. “In this context, health literacy refers to the ‘degree to which individuals have the capacity to obtain, process and understand basic health information and services needed to make appropriate health decisions’ [63].”

Is this then a convenience or a purposive sample? Line 430 has been amended to “A new cross-sectional purposive critical case sample” and reference provided. New reference added to list. 

Etikan I, Musa SA, Alkassim RS. Comparison of convenience sampling and purposive sampling. Am J Theor Appl Stat. 2016;5(1):1-4. doi: 10.11648/j.ajtas.20160501.11.

4. Have the authors described where all data underlying the findings will be made available when the study is complete?

Reviewer #1: No

Reviewer #2: No

The following statement has been added to the results section. 

“Availability of data and materials

The datasets generated and/or analysed during the study will not be made publicly available as restrictions apply to the availability of these data (intention of data analysis included in participant information forms) and sensitivity (i.e. human data, Māori data sovereignty) but will be available from the corresponding author on reasonable request. Data will be located in a controlled access repository at the University of Auckland.”

---

## [Decision Letter · Decision Letter 1]

24 Jun 2022

Health equity and wellbeing among older people’s caregivers in New Zealand during COVID-19: Protocol for a qualitative study

PONE-D-21-26434R1

Dear Dr. Burholt,

We’re pleased to inform you that your manuscript has been judged scientifically suitable for publication and will be formally accepted for publication once it meets all outstanding technical requirements.

Kind regards,

Maw Pin Tan, M.D.

Academic Editor

PLOS ONE

Additional Editor Comments (optional):

Reviewers' comments:

Reviewer's Responses to Questions

**Comments to the Author**

1. Does the manuscript provide a valid rationale for the proposed study, with clearly identified and justified research questions?

Reviewer #1: Yes

Reviewer #2: Yes

2. Is the protocol technically sound and planned in a manner that will lead to a meaningful outcome and allow testing the stated hypotheses?

Reviewer #1: Yes

Reviewer #2: Yes

3. Is the methodology feasible and described in sufficient detail to allow the work to be replicable?

Reviewer #1: Yes

Reviewer #2: Yes

4. Have the authors described where all data underlying the findings will be made available when the study is complete?

Reviewer #1: No

Reviewer #2: Yes

5. Is the manuscript presented in an intelligible fashion and written in standard English?

Reviewer #1: Yes

Reviewer #2: Yes

6. Review Comments to the Author

You may also provide optional suggestions and comments to authors that they might find helpful in planning their study.

Reviewer #1: The authors have addressed all the comments. I have no further comments. The manuscript appears ready for publication.

Reviewer #2: Revisions generally conducted in response to requests ------------------------------------------------

7. PLOS authors have the option to publish the peer review history of their article (what does this mean?). If published, this will include your full peer review and any attached files.

Reviewer #1: **Yes: **Sridhar Vaitheswaran

Reviewer #2: No

---

## [Editor Report · Acceptance letter]

7 Jul 2022

PONE-D-21-26434R1 

Health equity and wellbeing among older people’s caregivers in New Zealand during COVID-19: Protocol for a qualitative study 

Dear Dr. Burholt:

I'm pleased to inform you that your manuscript has been deemed suitable for publication in PLOS ONE. Congratulations! Your manuscript is now with our production department. 

Kind regards, 

on behalf of

Dr. Maw Pin Tan 

Academic Editor

PLOS ONE